# Identification of Potential miRNA-mRNA Regulatory Network Associated with Regulating Immunity and Metabolism in Pigs Induced by ASFV Infection

**DOI:** 10.3390/ani13071246

**Published:** 2023-04-04

**Authors:** Zhongbao Pang, Shiyu Chen, Shuai Cui, Wenzhu Zhai, Ying Huang, Xintao Gao, Yang Wang, Fei Jiang, Xiaoyu Guo, Yuxin Hao, Wencai Li, Lei Wang, Hongfei Zhu, Jiajun Wu, Hong Jia

**Affiliations:** 1Institute of Animal Sciences, Chinese Academy of Agricultural Sciences, Beijing 100081, China; 82101205453@caas.cn (Z.P.);; 2College of Animal Medicine, Shandong Vocational Animal Science and Veterinary College, Weifang 261061, China; 3Biotechnology Research Institute, Chinese Academy of Agricultural Sciences, Beijing 100081, China; 4China Animal Disease Control Center, Beijing 100026, China

**Keywords:** African swine fever virus, miRNA, GO function enrichment analysis, miRNA-mRNA network, KEGG signal pathway analysis

## Abstract

**Simple Summary:**

In order to understand the complex interactions between ASFV and the host cell, the expression differences and functional analysis of microRNA (miRNA) in porcine peripheral blood lymphocytes of ASFV infected pigs and healthy pigs were compared, and the regulatory network of miRNA-mRNA was drawn. miRNAs in pathways related to disease, inflammation, and lipid metabolism were differentially expressed. ASFV could regulate immunity and metabolism-related pathways in infected pigs by inducing differential expression of miRNAs. This study elucidated the immunity regulation mechanisms of ASFV, which will provide a theoretical basis for the development of good prevention and control strategies.

**Abstract:**

African swine fever (ASF) is a devastating infectious disease in domestic pigs caused by African swine fever virus (ASFV) with a mortality rate of about 100%. However, the understanding of the interaction between ASFV and host is still not clear. In this study, the expression differences and functional analysis of microRNA (miRNA) in porcine peripheral blood lymphocytes of ASFV infected pigs and healthy pigs were compared based on Illumina high-throughput sequencing, then the GO and KEGG signal pathways were analyzed. The miRNA related to immunity and inflammation were screened, and the regulatory network of miRNA-mRNA was drawn. A total of 70 differentially expressed miRNAs were found (*p* ≤ 0.05). Of these, 45 were upregulated and 25 were downregulated in ASFV-infected pigs vs. healthy pigs. A total of 8179 mRNA genes targeted by these 70 differentially expressed miRNA were predicted, of which 1447 mRNA genes were targeted by ssc-miR-2320-5p. Five differentially expressed miRNA were validated by RT-qPCR, which were consistent with the RNA-Seq results. The GO analysis revealed that a total of 30 gene functions were significantly enriched, including 7 molecular functions (MF), 13 cellular components (CC), and 10 biological processes (BP). The KEGG enrichment analysis revealed that the differentially expressed genes were significantly enriched in pathways related to immunity, inflammation, and various metabolic processes, in which a total of two downregulated miRNAs after infection and eight upregulated miRNAs related to immunity and inflammation were screened in ASFV-infected pigs vs. healthy pigs. The network of miRNA-mRNA showed that the mRNA target genes were strongly regulated by ssc-miR-214, ssc-miR-199b-3p, and ssc-miR-199a-3p. The mRNA target genes were enriched into the MAPK signaling pathway, Toll-like receptor signaling pathway, TNF signaling pathway, and IL-17 signaling pathway by using a KEGG enrichment analysis. Therefore, ASFV could regulate immunity and metabolism-related pathways in infected pigs by inducing differential expression of miRNAs. These results provided a new basis for further elucidating the interactions between ASFV and the host as well as the immunity regulation mechanisms of ASFV, which will be conducive to better controlling ASF.

## 1. Introduction

African swine fever (ASF) is an acute, febrile, and hemorrhagic disease in domestic and wild pigs that is caused by the African swine fever virus (ASFV). ASFV is a large-diameter icosahedral DNA virus, and the genome is a linear 170–190 kb double-stranded DNA [1]encoding 150–200 viral proteins, including 68 structural proteins and more than 100 non-structural proteins [2]; however, the functions of most proteins are still unknown [3,4]. ASF was first reported in Kenya in 1921 [5], and it subsequently became endemic in most of sub-Saharan Africa and Sardinia [6]. ASFV spread outside Africa to the Iberian Peninsula, initially to Portugal in 1957 and 1960, and subsequently to Europe and Latin America, then it was eradicated from all Western Europe except Sardinia by 1995 [7]. In 2007, ASF caused a serious outbreaks in Georgia, subsequently spreading to other adjacent countries of Europe, including Ukraine and Russia [1,8]. In 2018, the first outbreak of ASF was reported in Liaoning, China [9]. Infected pigs present with high fever, hemorrhage, and multiorgan dysfunction. Clinical signs appear 7–10 days after experimental infection, and the animals die shortly thereafter [10]. Monocyte–macrophages are the main target cells of the ASFV [11], and the ASFV has evolved several mechanisms to evade and suppress immune response, including blocking signal delivery on the antagonized cGAS/STING pathway [12], inhibiting apoptosis pathways [13], inhibiting major histocompatibility complex (MHC)-I/II expression, and cytotoxic T-cell activation [14]. Most of the molecular mechanism of ASFV infection in host cells is still unclear. The complex viral genome and its sophisticated ability to regulate the host immune response have seriously hindered the development of effective vaccines. So to date, there are still no effective vaccines or antiviral drugs for the prevention or treatment of ASF [15,16,17].

MicroRNAs (miRNAs) are small, non-coding RNA molecules that regulate gene expression after transcription of their coding genes. The transcripts are sheared by endonucleases, modified, and processed into pre-miRNA. Then, they are transported to the cytoplasm by exportin-5 [18,19] and processed by the enzyme Dicer to produce a mature miRNA of 18–23 nt [20]. miRNAs play a key regulatory role in various biological processes in vivo, including cell proliferation, differentiation, apoptosis, cancer, and immune regulation, and they have been implicated in cancer and other diseases [21]. Recently, important progress has been made in the study of non-coding RNAs, including miRNAs, in the pathogenesis of ASF. Crystal Jaing et al. compared the gene expression differences of whole-blood RNA from pigs infected with a low pathogenic ASFV isolate, OUR T88/3 (OURT), and the highly pathogenic Georgia 2007/1. Host genes associated with macrophages, those linked with virus infection, lymphocyte-associated genes with an emphasis on NK cells, and genes not associated with immunity such as TGM3 and GPATCH4 could be directly or indirectly associated with the response to infection with a highly pathogenic ASFV GRG2007/1. RNA-Seq revealed only a limited number of miRNA genes upregulated during ASFV infection, including miR-122, miR-138, miR-181A, miR-199A-2, and miR-1296. Potential roles for miRNAs in ASFV infection have not been investigated; miRNAs are likely involved in the regulation of virus replication and host responses as in other viral infections. A large percentage of genes were identified as downregulated, but the most highly downregulated genes could not be linked to virus infection, inflammation, or lymphocyte activation [3]. Whole-transcriptome RNA-Seq analyses were conducted in porcine alveolar macrophages (PAMs) infected with Pig/Heilongjiang/2018 (Pig/HLJ/18) ASFV. The results suggested a strong inhibition of host immunity-related genes by ASFV infection in PAMs, while enhanced chemokine-mediated signaling pathways and neutrophil chemotaxis were observed in ASFV-infected PAMs. Furthermore, ASFV infection also downregulated host microRNAs (miRNAs) that putatively targeted viral genes while also triggering dysregulation of the host metabolism, which promoted virus replication at the transcription level [22]. However, this study was conducted in cell culture; relatively few were conducted in vivo [10]. In this study, the expression differences and functional analysis of microRNA (miRNA) in porcine peripheral blood lymphocytes of ASFV-infected pigs and healthy pigs was compared. The differentially expressed genes were significantly enriched in pathways related to immunity, inflammation, and various metabolic processes. The mRNA target genes were strongly regulated by ssc-miR-214, ssc-miR-199b-3p, and ssc-miR-199a-3p. The mRNA target genes were enriched into the MAPK signaling pathway, Toll-like receptor signaling pathway, TNF signaling pathway, and IL-17 signaling pathway. So, ASFV could regulate immunity and metabolism-related pathways in infected pigs by inducing differential expression of miRNAs. This is helpful in elucidating the immunity regulation mechanisms of ASFV, which will provide a theoretical basis for effectively controlling ASF by taking advantage of the sncRNA system.

## 2. Materials and Methods

### 2.1. Sample Collection

Six 4-week-old British Large White pigs were obtained from Beijing SPF Pig Breeding Management Center that were negative for ASFV, porcine reproductive and respiratory syndrome virus (PRRSV), porcine circovirus (PCV), and pseudorabies virus (PRV) as tested by PCR and ELISA. The pigs were randomly divided into two groups and housed in the ABSL-3 laboratory of China Animal Disease Control Center (CADC), then were infected with 10^2^ HAD_50_ of the ASFV CADC_HN09 strain provided by CADC or PBS, respectively. In order to detect the infection status of these pigs, the clinical symptoms and body temperature changes were recorded every day, then the oral, nasal, and anal swabs and serum were collected at 7 DPI to detect the virus load, and the PBLs were isolated from the 5 mL of anticoagulant blood using a Solabio LIFE SCIENCES kit (P8770). All PBL samples were stored at −80 °C until use. To quantify the ASFV load in different samples, the total genomic DNA was extracted from swabs and serum to detect the ASFV B646L gene (VP72-F1:5′-GCTTTCAGGATAGAGATACAGCTCT-3′, VP72-R1:5′-CCGTAGTGGAAGGGTATGTAAGAG-3′ and VP72- probe: FAM-CCGTAACTGCTCGTATCAATCTTATCG-BHQ1), and the serum was detected using commercial kits (IDVet Co., Ltd., Grabels, France) according to the instructions.

### 2.2. RNA Extraction and Quality Detection

All PBL samples from the infected group and control group were detected by high-throughput sequencing. The total sample RNA was extracted using TRIzol^®^ Reagent (Invitrogen Co., Ltd., Waltham, MA, USA) following the manufacturer’s protocol, and the RNA obtained was assessed for sample integrity via 1% agarose gel electrophoresis and for concentration and purity using a NanoDrop2000. We considered 28S/18S = 1.8–2.2 and RIN > 7 as an indication of good RNA integrity, and an RNA concentration ≥50 ng/μL with a total amount not less than 1 μg was found to be sufficient for library construction.

### 2.3. Establishment of an sRNA Sequencing Library and Sequencing

After RNA extraction and quality assessment, libraries were constructed using 1 μg of total RNA. The 3′ and 5′ ends of the RNA were ligated using a TruSeq Small RNA sample prep kit (Illumina, San Diego, CA, USA), the RNA was reversed-transcribed into cDNA using random primers, and the library was enriched by PCR amplification. The product library was purified via 6% polyacrylamide gel electrophoresis. Then, bridge PCR amplification was performed using a cBot system to generate clusters, and SE50 sequencing was performed on a Hiseq sequencing platform (Illumina, San Diego, CA, USA).

### 2.4. Bioinformatic Analysis

Raw reads generated by sequencing were quality-controlled using Fastx-Toolkit software to remove low-quality bases, linker sequences, reads containing more than 10% N, and short reads to obtain high-quality clean reads for subsequent bioinformatic analyses. The Rfam (11.0, http:Rfam.sanger.ac.uk (accessed on 10 March 2022)) database was used to annotate small RNA; identify and remove non-miRNA sequences such as rRNA, scRNA, snoRNA, snRNA, and tRNA; and perform statistics on the species and number of small RNAs present. Small RNAs were compared to the pig reference genome using Bowtie 1.2.1.1 (https://www.ncbi.nlm.nih.gov/genome/?term=txid9822[Organism:exp (accessed on 10 March 2022)) to determine their locations in the porcine reference genome. Additionally, expression of recognized miRNAs was quantified and normalized using transcripts per million (TPM) reads as follows:TPM = miR − read counts × 1,000,000/library size

Differences in miRNA expression between samples from the control and infected groups were identified, and a pattern clustering analysis was performed to determine the differentially expressed genes.

### 2.5. Target Gene Prediction and Enrichment of Differentially Expressed miRNA

The target genes of differentially expressed miRNA were predicted using Miranda software. GO and KEGG enrichment analyses of the target genes were achieved using GO (http://geneontology.org (accessed on 12 March 2022)) and KEGG enrichment software (https://www.kegg.jp (accessed on 12 March 2022)), respectively, and *p*-values were corrected using Fisher’s exact test and four multiple test methods (Bonferroni, Holm, Sidak, and false discovery rate). Expression was considered to be significantly enriched when the *p*-value was not greater than 0.05.

### 2.6. Combined Analysis of Differentially Expressed miRNAs and Target Gene mRNA

According to the results of miRNA sequencing and mRNA sequencing, the differentially expressed miRNA and mRNA of ASFV CADC_HN09-infected and control groups were obtained, and the miRNA-mRNA combined analysis was carried out in order to screen out the miRNA related to host immunity or inflammation after virus infection. MiRNA is a negative regulatory factor that can negatively regulate the expression of its downstream target genes. In this study, the enrichment of signal pathways and the regulatory pathways involved in differentially expressed genes were analyzed using Cytoscape software to map the regulatory network.

### 2.7. Verification of Differentially Expressed miRNA by Real-Time Quantitative Polymerase Chain Reaction (qPCR)

To validate the sequencing results, qPCR was performed using miRNA-specific primers. Five miRNAs were randomly screened using the miR-16 gene as the internal reference and three biological replicates performed per sample. Using AceQ qPCR SYBR Green Master Mix (Jizhen Biology, Q121-02), the system and procedure were as follows: cDNA was synthesized using a HiScript 1st Strand cDNA Synthesis Kit (Vazyme Biotech Co., Ltd., AORT-0060) with the following reaction procedure: 25 °C for 5 min, 50 °C for 15 min, and 85 °C for 5 min. Then, 2× RT-PCR Mix 10 μL, HiScript Enzyme Mix 2 μL, Oligo (dT) 18 (50 M) 1 μL, random hexamers (50 ng/μL) 1 μL, total RNA 1 μL, DEPC H2O 5 μL: 25 °C for 5 min, 50 °C for 15 min, and 85 °C for 5 min. In step 2, using the AceQ qPCR SYBR Green Master Mix to perform qRT-PCR, 2× SYBR Green Mix 5 μL, Primer F+ R (each 10 μM) 0.5 μL, cDNA 2 μL, and DEPC water 2 μL were used: preheating at 95 °C for 5 min, 40 cycles of 95 °C for 10 s and 60 °C annealing/extension for 30 s, dissolution curve of 95 °C for 15 s, 60 °C for 60 s, 95 °C for 30 s, and 95 °C for 15 s. The miRNA was detected using the stem-loop method. The upstream primer sequences are shown in Table 1. The downstream primer was a universal primer (sequence = AGTGCAGGTCCGAGGTATT) provided in the reverse transcription.

## 3. Results

### 3.1. Clinical Sample Analysis via qPCR and ELISA

All pigs infected with 10^2^ HAD_50_ of ASFV CADC_HN09 presented with a rise in body temperature (above 40.5 °C) from day 3 dpi (Figure 1), which quickly evolved to full clinical disease (depression, anorexia, staggering gait, diarrhea, and purple skin discoloration), among which two pigs died at 7 dpi, and all died at 8 dpi. The antibodies were negative in serum of both ASFV-infected pigs and healthy pigs as detected by indirect ELISA (S/P% ≤ 30%) (Figure 2A) and competition ELISA (S/N% ≥ 50%) (Figure 2B). To quantify the ASFV load in different samples, qPCR was performed to detect the P72 gene. The results showed that ASFV was detected in blood and swab samples from infected pigs but was not detected in samples of healthy pigs, and the highest levels of viral DNA were detected in the blood and nasal samples of infected pigs (Figure 2C).

### 3.2. Establishment of an sRNA Sequencing Library and Sequencing

Total RNA of all PBL samples from the infected group and control group were extracted, and the libraries were constructed and detected by Illumina high-throughput sequencing. Raw reads were obtained following high-throughput sequencing of these samples. As shown in Table 2, after removing the low-quality and contaminated sequences from the original library, 39,641,228 pure sequences of 18–32 nt were obtained. The analysis showed that most sRNAs were 21–23 nt in length, accounting for 60% of all sRNAs detected (Figure 3), and that their sizes were not normally distributed, with 22 nt representing 31% of all sRNAs and occurring most frequently. The results of sequencing six libraries indicated that the GC content of these sRNAs was >60% and their AT content was <GC. Additionally, Q20 and Q30 were greater than 86% and 80%, respectively, and the base error rate was low. 

Next, the first-base composition of miRNAs, a subset of the sRNAs, with different lengths was determined (Figure 4). For sRNAs with a length of 22–24 nt, the first bases from the control and infected groups were significantly different. The first bases of sRNAs from the control group were predominantly guanine (G), whereas the first bases of sRNAs from the infected group were only 22 nt in favor of G. The first bases of sRNAs > 24 nt were predominantly uracil (U), whereas the first bases of sRNAs < 22 nt were dominated by cytosine (C) and G. The majority of sRNAs were 21–23 nt in length. A comprehensive analysis revealed that the first bases of miRNA in all samples had a bias toward G and U.

### 3.3. Database Comparison

The detected sRNAs were annotated, and non-miRNA sequences were removed using information from the Rfam and Bowtie databases. Then, mRNAs were localized to the porcine reference genome, and the locations of the sequences on the genome were determined (Figure 5). There were 34,952,559 miRNAs located in the reference genome, accounting for 88.2% of the clean reads (34,952,559/39,641,228). These sequences were mainly found on chromosomes 5, 6, 12, and 18 and the X chromosome; chromosome 8 and the Y and MT chromosomes had very few miRNA sequences. In terms of strand preference, sequences mapping to chromosomes 13 and 18 were predominantly found on positive strands, while those on chromosomes 6, 9, and 10 and the X chromosome were mainly found on negative strands.

### 3.4. miRNAs Differentially Expressed in the Transcriptome

The probability of statistical hypothesis testing (*p*-value) and Benjamini and Hochberg correction (BH) were calculated after normalizing the read count. A total of 70 differentially expressed genes were identified. Among them, 45 were upregulated and 25 were downregulated. Due to the large number of differentially expressed genes, only miRNAs with a log^2^ fold-change ≥ 4 or ≤ −2 are listed (Table 3). Across these genes, the fold change and number of genes upregulated were significantly more than those downregulated, and downregulated genes showed an approximately 2-fold change. Further, ssc-miR-122 was found to be negligibly expressed in the control group, while it was significantly upregulated expression in the infected group, indicating that this miRNA might play a specific role after ASFV infection.

### 3.5. Target Gene Prediction and Function Enrichment Analyses

For the differentially expressed miRNAs, target genes were predicted using the Miranda program. A total of 22,890 target genes were predicted for the 70 differentially expressed miRNAs with different transcripts. This number of the target genes was reduced to 8179 after de-duplication, suggesting that multiple miRNAs could regulate the same gene and the same miRNA could regulate multiple genes. For example, TRAF3 was co-regulated by multiple miRNAs, including ssc-miR-139-3p, ssc-miR-1306-5p, ssc-miR-2320-5p, and ssc-miR-339-3p. GO functional and KEGG pathway analyses were performed on the target genes based on the differentially expressed genes/transcripts identified. The GO analysis revealed that a total of 30 gene functions were significantly enriched, including 7 molecular functions (MF) mainly included catalytic activity, protein binding and kinase activity; 13 cellular components (CC) mainly included cell, cytoplasm, and organelle parts; and 10 biological processes (BP) mainly included in regulation of metabolic process, cellular metabolic processes, and intracellular signal transduction (Figure 6). The KEGG enrichment analysis revealed that the differentially expressed genes were significantly enriched in pathways related to immunity, inflammation, and various metabolic processes, including those associated with leukocyte transendothelial migration, Toll and Imd signaling, influenza A, and lysosomes. The MAPK signaling pathway, which is related to immunity, as well as the fatty acid metabolism, secondary product metabolism, fatty acid biosynthesis, and estrogen signaling pathway pathways, which are involved in lipid metabolism, were also enriched in genes that were differentially expressed (Figure 7).

### 3.6. Combined Analysis of Differentially Expressed miRNAs and Target Gene mRNA

According to the results of miRNA sequencing, the differentially expressed miRNA of the ASFV CADC_HN09-infected and control groups were obtained, and total of two downregulated miRNAs (Table 4) and eight upregulated (Table 5) miRNAs after infection related to immunity and inflammation were screened. The miRNA-mRNA combined analysis and regulatory network map was constructed using Cytoscape software (Figure 8). The network of miRNA-mRNA showed that ssc-miR-214, ssc-miR-199b-3p, and ssc-miR-199a-3p had strong regulatory effects on the target genes. The target genes were enriched into four signal pathways using a KEGG enrichment analysis (Table 6).

### 3.7. Verification of Differentially Expressed miRNA by qPCR

To verify the accuracy of sequencing results, five differentially expressed genes—miR-362, miR-744, miR-486, miR-139-3p, and miR-199a-5P—were randomly selected for qPCR validation. Results of the qPCR and RNA-Seq were generally consistent (Figure 9).

## 4. Discussion

Since its first appearance in China in 2018, ASF has caused heavy losses and has seriously hindered the development of the pig industry. The development of safe and effective vaccines to protect pigs against ASF has been hindered by a lack of understanding of the complex interactions between ASFV and the host cell. As a small-molecule RNA, miRNAs bind to complementary sequences in the 3’-UTR of target genes, leading to degradation of target genes and inhibition of gene translation. miRNAs play an important role in the response to viral infections and immune regulation. The prediction of miRNA target genes is valuable for understanding the functions and targets of miRNA molecules, and bio-functional analyses can further help us understand the pathways and gene processes they regulate.

Differential expression of host genes after viral infection has attracted substantial attention. However, most studies have been performed in vitro, and there are relatively few reported studies conducted in vivo. In this study, lymphocytes in the peripheral blood of ASFV-infected pigs were selected for the RNA-Seq analysis. Sequencing of mRNAs from the PBLs of each group (control and infected groups) was accomplished using the Illumina platform, and approximately 40,000,000 clean sequences were obtained with an error rate of less than 0.5%. Almost 90% of these reads mapped to the reference genome, indicating that the sequencing data were of good quality and high accuracy. 

In this study, target gene prediction and functional annotation of mRNA genes differentially expressed in pigs in the control and infected groups revealed that target genes of these miRNA were mainly involved in two major signaling pathways. First, these miRNAs acted on metabolic pathways, including those for fatty acid metabolism, secondary product metabolism, estrogen signaling, and insulin signaling. Indeed, insulin signaling plays an important role in the regulation of adipogenesis [23,24]. As an endocrine tissue, adipose secretes a large number of bioactive molecules and plays an important role in vascular function, immune regulation, and adipocyte metabolism [25]. Previously, cholesterol metabolism was shown to be closely related to ASFV infection and replication of the virus in host cells. miR-10b specifically showed differential expression within a very short period of time after viral infection, in which it targeted the ATP binding cassette transporter A1 to regulate cholesterol efflux from host cells [26]. Similarly, as a steroid hormone, estrogen might also play a role in this process. Second, these miRNA acted on pathways related to disease and inflammation, including those for leukocyte transendothelial migration, Toll and Imd signaling, influenza A, lysosome, and endophagy. Indeed, lysosomes cooperate with phagosomes during autophagy to remove debris; they also break down virus particles or bacteria engulfed by macrophages [27], which might be related to the entry or release of virus particles immediately after infection, similar to the mechanisms of virus entry and cell apoptosis during the peak period of virus replication [10].

miRNAs have been shown promote the replication of PRRSV, with the virus upregulating miR-373 by regulating the expression of Sp1, which acts on target genes (e.g., IRAK1, IRF1, and NFI) to inhibit β-interferon production, thereby promoting virus replication in vitro [28]. Tongcheng-specific or Landrace-specific DEmiRNAs might reflect breed-specific antiviral mechanisms. Li et al. [29] compared Tongcheng pigs before and after PRRSV infection to find Tongcheng-specific DEmiRNA-22-5p was significantly upregulated at all the time points.

Analysis of differentially expressed miRNAs revealed negligible or no expression of five miRNAs (ssc-miR-9858-5p, ssc-miR-122, ssc-miR-2366, ssc-miR-145-5p, and ssc-miR-214) from the control group but upregulated expression in the infected group, suggesting that these miRNAs might play key roles during viral infection. miR-22 was reported to significantly suppress the activity of NF-kB by regulating the expression of nuclear receptor coactivator 1 (NCOA1) [30]. Type I interferon plays a key role in antiviral responses (as well as in the ASFV infection process), with genes in the variable region of the virus inhibiting the production of type I interferon and regulating the expression of proinflammatory cytokines [31]. The targeted prediction of miR-122 suggests that it interacts with multigene families, replication, unknown function and unknown genes, and multigene family members modulate host innate responses by determining tropism, virulence, and inhibition of type I IFN responses [10,32]. The PI3K-Akt signaling pathway was also enriched in the targets of these miRNAs detected in our studies. Our results were similar to previous studies that showed that miR-122 is involved in multiple immune signaling pathways, including those related to T-cell and B-cell receptor signaling pathways [10]. Liu et al. indicated that miR-122 overexpression appears to exacerbate the angiotensin II-mediated loss of autophagy and increased inflammation, apoptosis, extracellular matrix deposition, cardiovascular fibrosis, and dysfunction by modulating the SIRT6-Elabela-ACE2, LGR4-β-catenin, TGFβ-CTGF, and PTEN-PI3K-Akt signaling pathways. More importantly, the inhibition of miR-122 has proautophagic, antioxidant, anti-inflammatory, antiapoptotic, and antifibrotic effects [33]. ASFV infection can activate the RLR and TLR signaling pathways, while TLR4 and TLR6 are significantly downregulated after infection. miR-21 and miR-145-5p regulate the expression of TLR4 by activating the TLR signaling pathway [34]. Activation of MyD88-dependent and/or MyD88-independent pathways induces TLR4 inflammatory responses, which in turn promote the expression of proinflammatory transcription factors such as nuclear factor-κB (NF-κB) [35], which plays a crucial role in the induction of inflammatory mediators (cytokines, chemokines, or co-stimulatory molecules) associated with the development and progression of many chronic diseases. Both miR-451 and miR-145-5p were expressed at high levels in spleen tissues after infection with ASFV E75 strains and were upregulated at 7 dpi and downregulated at 3 dpi [10], consistent with our results at 7 dpi. With miR-451, the most representative miRNA in the spleen of virulence-infected animals, overexpression at 7 dpi may help to inhibit autophagy to promote ASFV replication and avoid viral clearance [36]. In vitro cell-level studies found that miR-10b expression was significantly upregulated in a very short time after ASFV infection [29], but we still found that this miRNA was significantly upregulated at 7 d of infection, so acting on the target ABCA1 may play an important role in the early infection of ASFV mediated by macrophage endocytosis and viral replication.

miRNA-mRNA regulatory networks have been shown to regulate multiple biological pathways and processes by means of complex relationships [37]. In this study, a total of two downregulated miRNAs (miR-450c-5p and miR-374b-3p) and eight upregulated miRNAs (miR-9858-5p, miR-195, miR-122, miR-199b-3p, miR-199a-3p, miR-10b, miR-145-5p, and miR-214) after infection with ASFV CADC_HN09 related to immunity and inflammation were screened. The network of miRNA-mRNA showed that ssc-miR-214, ssc-miR-199b-3p, and ssc-miR-199a-3p had strong regulatory effects on the target genes. The target genes were enriched into the MAPK signaling pathway, Toll-like receptor signaling pathway, TNF signaling pathway, and IL-17 signaling pathway using a KEGG enrichment analysis. All of these pathways regulate immunity functions of infected hosts.

In summary, our results were similar to those of previous studies and indicated that the above miRNAs might play important roles in immune inflammation as well as in cellular autophagy, apoptosis, and other processes. All pigs infected with 10^2^ HAD_50_ of ASFV CADC_HN09 presented with a rise in body temperature (above 40.5 °C) from day 3 dpi, which quickly evolved to depression, anorexia, staggering gait, diarrhea, and purple skin discoloration. In addition, fever (>41 °C) lasted for 3 days; and anorexia, lameness, dyspnea, bloody diarrhea and cyanosis appeared at 7 dpi (two pigs died at 7 dpi, and all died at 8 dpi). In order to compare the most significant differences, microRNA (miRNA) in porcine peripheral blood lymphocytes of ASFV-infected pigs and healthy pigs at 7 dpi was compared based on Illumina high-throughput sequencing. However, the differences in microRNA (miRNA) at 7 dpi could not fully reflect all the differences caused by ASFV infection. Therefore, in future research, the differences in miRNA at different time points need to be further investigated and verified so as to more comprehensively reflect the impact of ASFV infection on the expression of miRNA in the host.

## 5. Conclusions

In summary, 70 miRNAs differentially expressed in pigs in response to ASFV infection were identified by high-throughput sequencing. Target gene prediction and annotation showed that these miRNAs were significantly enriched in pathways related to disease, inflammation, and lipid metabolism. The results of this study expand our understanding of miRNA expression during ASFV infection. In addition, these findings can help elucidate virus infection mechanisms and virus–host interactions as well as provide a theoretical basis for the development of good prevention and control strategies.

## Figures and Tables

**Figure 1 animals-13-01246-f001:**
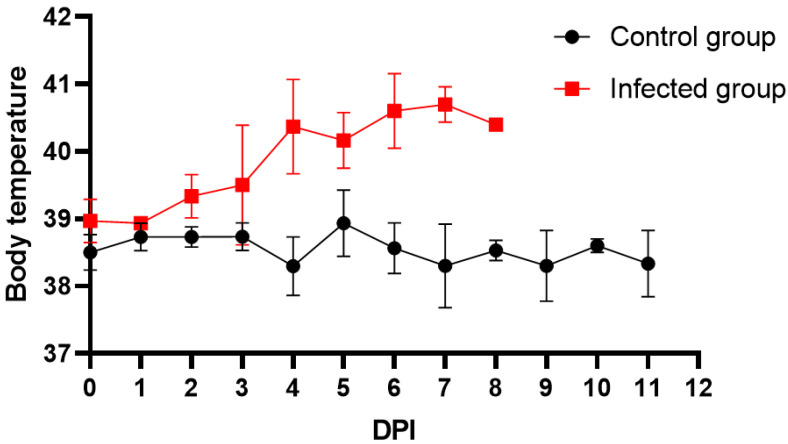
Body temperature detection. Six 4-week-old pigs were randomly divided into two groups and then infected with 10^2^ HAD_50_ of the ASFV CADC_HN09 strain and PBS, respectively. All infected pigs presented with a rise in body temperature (above 40.5 °C) from day 3 dpi, and 2 piglets died at 7 dpi, and all died at 8 dpi.

**Figure 2 animals-13-01246-f002:**
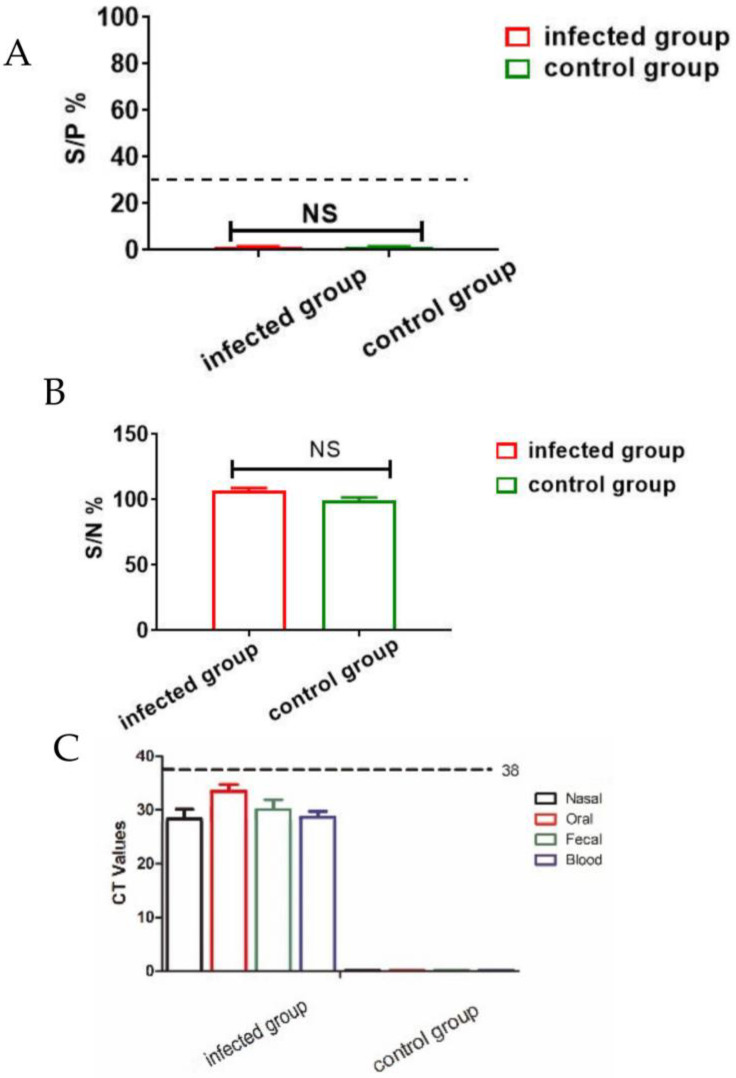
Clinical sample analysis by ELISA and qPCR. (**A**,**B**) Analysis of ASFV antibody levels in serum samples from infected group and control group were detected by using indirect ELISA (S/P% ≤ 30%) (**A**) and competition ELISA (S/N% ≥ 50%) (**B**). NS: there is no significant difference between infected group and control group. (**C**) ASFV B646L (p72) gene detection in different samples by qPCR. DNA was extracted from nasal, oral, fecal and blood samples from infected group and control group. The error bars represent the standard deviation among replicates. Data are shown as the mean ± SD based on three independent experiments.

**Figure 3 animals-13-01246-f003:**
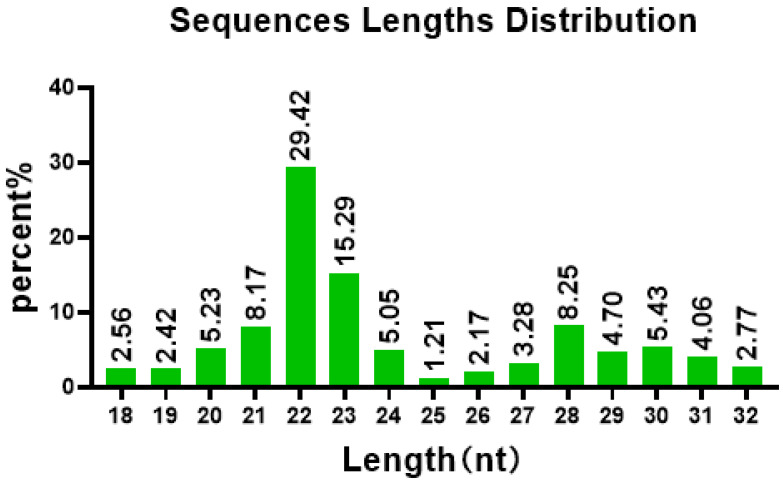
sRNA length distribution. The abscissa represents the miRNA length, and the ordinate represents proportion of miRNA at this length.

**Figure 4 animals-13-01246-f004:**
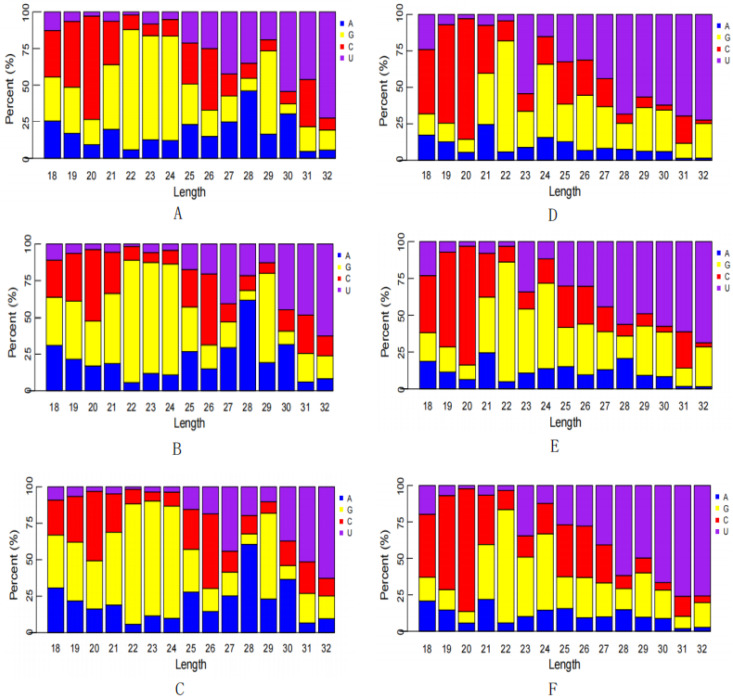
First base bias of miRNAs. (**A**–**C**) Control group; (**D**–**F**) infected group.

**Figure 5 animals-13-01246-f005:**
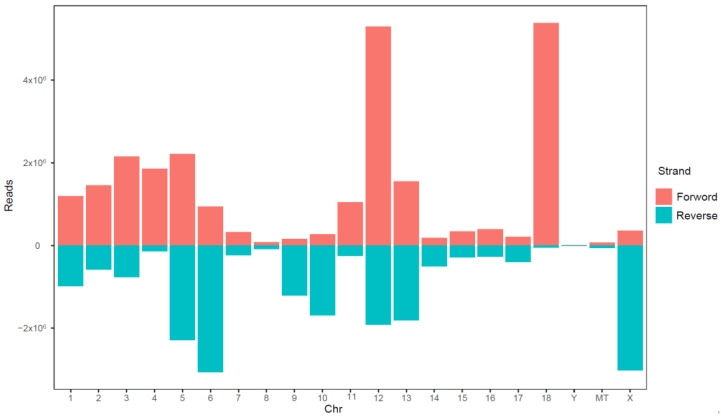
Distribution of miRNA coding sequences on positive and negative DNA strands. The abscissa represents chromosomal position, and the ordinate represents reads count.

**Figure 6 animals-13-01246-f006:**
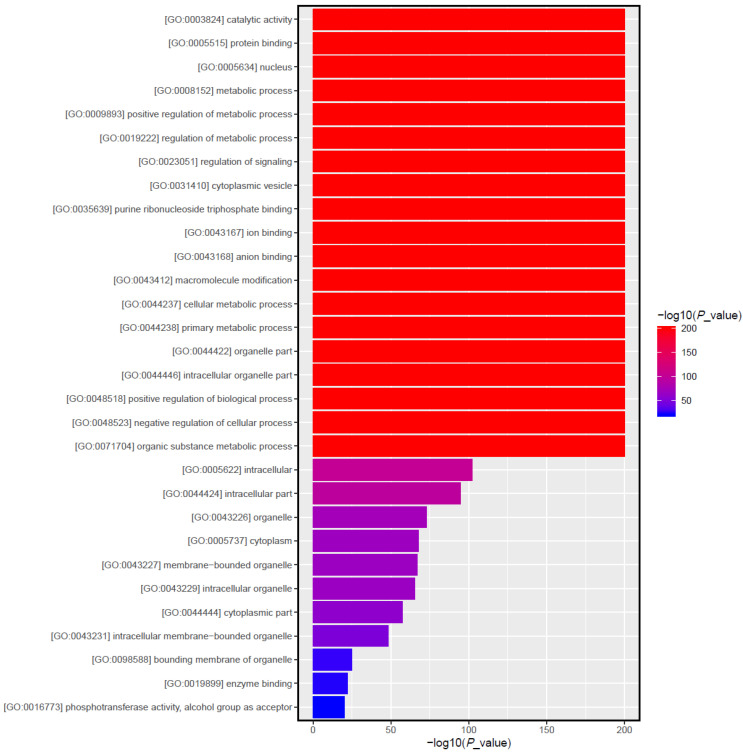
Functions associated with differentially expressed miRNA target genes identified by GO enrichment analysis. Bar graph of GO enrichment of differential genes intuitively reflects the distribution of the number of differential genes on GO term enriched in BP, CC, and MF. The 30 GO terms with the most significant enrichment were picked. Each column in the figure is a GO term, and the abscissa text indicates the name of the GO. The length of the column and the abscissa indicate the significance of the enrichment, which is the *p*-value; a darker color indicates a more significant enrichment of the function, and the color gradient on the right indicates the *p*-value size.

**Figure 7 animals-13-01246-f007:**
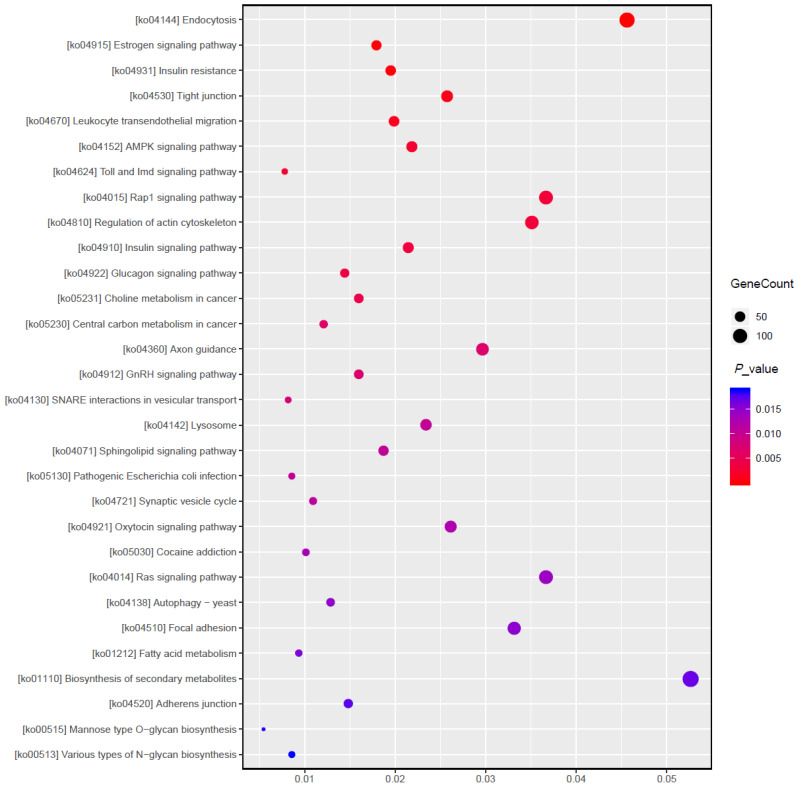
Pathways associated with differentially expressed miRNA target genes identified by KEGG enrichment analysis. Each point in this figure is a KEGG pathway, the ordinate text indicates the pathway name of KEGG, and the classification description is such as the Class legend information on the right. The abscissa is expressed as the enrichment rate, and the formula is as follows: (Enrich_factor = GeneRatio/BgRatio). A larger figure indicates a greater number of differentially expressed genes. The color indicates the significance of the enrichment, which is the *p*-value; a darker color indicates a more enriched pathway, and the color gradient on the right indicates the *p*-value size.

**Figure 8 animals-13-01246-f008:**
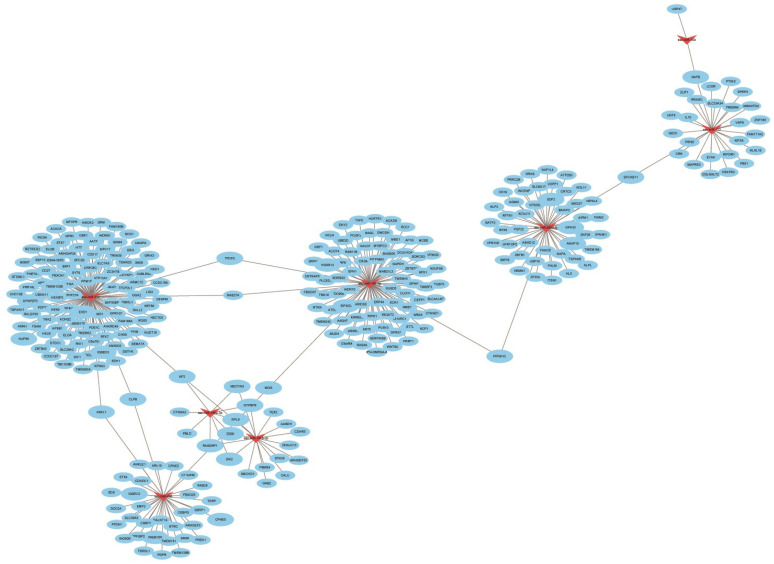
Regulatory network of differentially expressed miRNAs and mRNA of target genes. Red triangles represent miRNAs, and blue circles represent mRNAs.

**Figure 9 animals-13-01246-f009:**
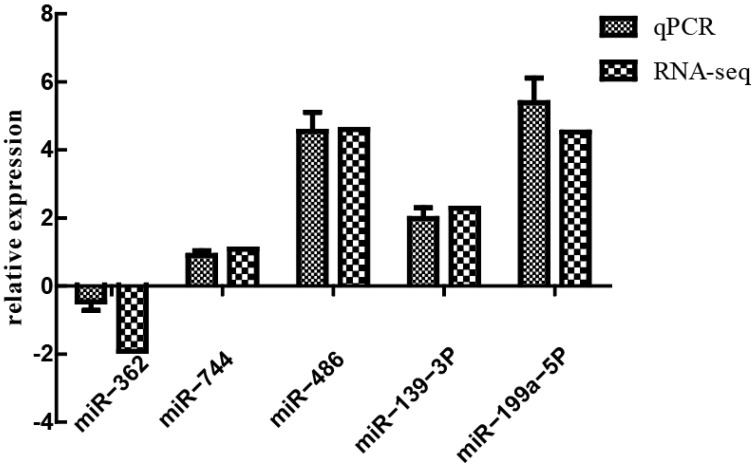
Relative expression of selected miRNA genes determined via qPCR.

**Table 1 animals-13-01246-t001:** Primer information.

miRNA		
	Primary Sequences (5′→3′)	Primer Sequences (5′→3′)
miR-362	AAUCCUUGGAACCUAGGUGUGAGUG	CGAATCCTTGGAACCTAGGTG
miR-744	UGCGGGGCUAGGGCUAACAGCA	GTGCGGGGCTAGGGCTA
miR-486	UCCUGUACUGAGCUGCCCCGAG	CGCGTCCTGTACTGAGCTGC
miR-139-3P	UGGAGACGCGGCCCUGUUGGAGU	GGAGACGCGGCCCTGT
miR-199a-5P	CCCAGUGUUCAGACUACCUGUUC	CGCGCCCAGTGTTCAGACTAC
miR-16	CCCAGUGUUCAGACUACCUGUUC	CGCGTAGCAGCACGTAAATA

**Table 2 animals-13-01246-t002:** sRNA high-throughput sequencing data.

Sample	Control Group			Infected Group		
	NC-1	NC-2	NC-3	YD-1	YD-2	YD-3
Total bases	1,427,067,450	1,231,831,950	3,334,317,150	3,180,968,400	3,126,361,800	4,273,526,700
Clean reads	3,526,712	3,820,877	10,695,589	10,430,439	9,873,282	12,949,329
Error %	0.0322	0.0393	0.0532	0.0524	0.0356	0.0333
Q20%	94.63	91.85	86.33	86.76	93.34	94.21
Q30%	88.95	85.02	80.45	80.5	86.83	88.23
GC%	62.41	62.67	64.17	67.43	67.41	67.22

**Table 3 animals-13-01246-t003:** Differentially expressed miRNAs.

miRNA	Control Group	Infected Group	log^2^ Fold Change (Control/Experimental)	*p*-Value	Up- or Downregulation
ssc-miR-486	1520.86	62.58	4.602820854	1.35 × 10^−20^	up
ssc-miR-451	4920.973333	309.1666667	3.992443281	3.20 × 10^−13^	up
ssc-miR-199a-5p	74.52666667	3.233333333	4.522401093	3.40 × 10^−7^	up
ssc-miR-199a-3p	116.51	6.91	4.073661759	8.58 × 10^−7^	up
ssc-miR-199b-3p	116.51	6.91	4.073661759	8.58 × 10^−7^	up
ssc-miR-122	205.08	0.053333333	11.66100452	1.42 × 10^−5^	up
ssc-miR-10b	528.52	22.75333333	4.537201625	0.000154438	up
ssc-miR-9858-5p	10.74666667	0.556666667	4.246586926	0.000620378	up
ssc-miR-2366	6.616666667	0.22	4.848577585	0.000771554	up
ssc-miR-145-5p	16.6	0.433333333	5.227516423	0.001947544	up
ssc-miR-214	6.63	0.22	4.851477475	0.008490924	up
ssc-miR-374a-3p	100.05	524.2333333	−2.389371253	3.63 × 10^−19^	down
ssc-miR-142-3p	1853.07	8993.783333	−2.279004567	2.74 × 10^−11^	down
ssc-miR-3613	4.34	24.9	−2.517637716	1.75 × 10^−6^	down
ssc-miR-374b-3p	3.536666667	18.32	−2.369671136	0.000189815	down
ssc-miR-450c-5p	26.29	120.34	−2.194101437	0.000543707	down

**Table 4 animals-13-01246-t004:** Target genes corresponding to the downregulated miRNA after infection.

Downregulated miRNA after Infection	Upregulation of mRNA
ssc-miR-450c-5p	GRIA3, GTF3C1
ssc-miR-374b-3p	HOOK2, STUM

**Table 5 animals-13-01246-t005:** Target genes corresponding to the upregulated miRNA after infection.

Upregulated miRNA after Infection	Downregulation of mRNA
ssc-miR-9858-5p	FAIM2, CDC25B, MAP2K4, IL17C, NOL11, SLC27A4, IL12RB2, BCL2L12, RPS15A
ssc-miR-195	MAP4, ABI2, MAPK10
ssc-miR-122	IL10, IFN-OMEGA-6, SLC8A3
ssc-miR-199b-3p	CAMK4, IRAG1, STAB1
ssc-miR-199a-3p	ARPC4
ssc-miR-10b	CCDC88C, THAP1, CCDC88C
ssc-miR-145-5p	JAM2, TRAF3, CD79B, SLC1A2, IL17RB, SLC44A2, IFT80, TLE3
ssc-miR-214	SLC7A1, IL12RB1, TNK2, TLCD4, IRF1, MAPK10, TLR4, TRIM59, IARS2, TRPM7, LRATD2

**Table 6 animals-13-01246-t006:** KEGG pathways enriched for target genes.

ID	Term	Counts	Gene Symbol
map04010	MAPK signaling pathway	2	CDC25B, MAP2K4
ko04620	Toll-like receptor signaling pathway	9	MAP2K4, MAPK10, TRAF3, IRF7, AK4, MAP3K8, TLR4, TLR8, TLR5
ko04668	TNF signaling pathway	4	NFKB1, TRAF3, TNF, IRF1
ko04657	IL-17 signaling pathway	3	IL17B, IL17C, NFKB1

## Data Availability

The data presented in this study are available in the manuscript.

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
