# Peer review of "Identification of Potential miRNA-mRNA Regulatory Network Associated with Regulating Immunity and Metabolism in Pigs Induced by ASFV Infection"

_animals, 2023, doi:10.3390/ani13071246_

Round 1

Reviewer 1 Report

The MS entitled “Identification of Potential miRNA–mRNA Regulatory Network Associated with Regulating Immunity and Metabolism in Pigs Induced by ASFV Infection” (animals-2238119) submitted by Pang et al is an interesting study about the immunity related to swine fever (ASF) . The MS is well planned, organized and well written. The Abstract is very informative and covered the entire work. The introduction is very informative and covered the addressed the problems. In methodology, internationally accepted protocols were followed and the data analyzed by advanced bioinformatics tools etc. The discussion part is very clear and discussed with appropriate references. The MS may accepting after clearing few typographical errors.       

Author Response

Dear Reviewer

Thanks for your letter and for the reviewer’ comments concerning our manuscript entitled “Identification of Potential miRNA–mRNA Regulatory Network Associated with Regulating Immunity and Metabolism in Pigs Induced by ASFV Infection”. Firstly, we would like to thank the reviewer and the editor for those comments which are all valuable and very helpful for revising and improving our paper. We have studied comments carefully and have substantially revised our manuscript which we hope to meet with editor and revisers’ standards. Revised portions are marked with red font in the paper.

Finally, we appreciate very much for your time in editing our manuscript and the valuable suggestions and comments. I am looking forward to hearing from your decision when it is made.

With best regards.

Reviewer 2 Report

The manuscript submitted by Pang et al. entitled "Identification of Potential miRNA–mRNA Regulatory Network Associated with Regulating Immunity and Metabolism in Pigs Induced by ASFV Infection" aims to improve the knowledge about the role of the miRNAs during the infection of ASFV in pigs. The results presented are novel, however the manuscript needs a deep editing revision. The text shows many mistakes, typos and errors that are not acceptable in a scientific article, in particular, in the abstract and introduction. In contrast, the M&M section and discussion sections are much more friendly to reader. Regarding the scientific aspects of the article, the authors report that 70 miRNAs are differentially expressed in pigs in response to ASFV infection being identified by high-throughput sequencing. Target gene prediction and annotation showed that these miRNAs were significantly enriched in pathways related to disease, inflammation, and lipid metabolism. Alltogether, the reported results deep our under
standing on the miRNA expression during ASFV infection, and help to elucidate the mechanisms of
infection and virus-host interactions.

These and many other mistakes and incorrections should be corrected be the acceptance of the manuscript for publication:

Line 19 - Explain: "The to protect pigs against ASF has been hindered by."

Line 23- Correct: "102 HAD50 ASFV" and check along the manuscript

Line 67- A statement on the absence of antivirals and vaccines should be added, as these two references (https://doi.org/10.1080/22221751.2022.2108342; https://doi.org/10.1080/22221751.2021.1902751).

Line 92 - Explain the rational of a single collection and at the 7dpi. Please discuss this limitation of the study in the discussion section.

Line 166-172: please rephrase the part of the manuscript and remove the description of M&M from here.

Author Response

Dear Reviewer

Thanks for your letter and for the reviewer’ comments concerning our manuscript entitled “Identification of Potential miRNA–mRNA Regulatory Network Associated with Regulating Immunity and Metabolism in Pigs Induced by ASFV Infection” (animals-2238119). Firstly, we would like to thank the reviewer for those comments which are all valuable and very helpful for revising and improving our paper. We have studied comments carefully and have substantially revised our manuscript which we hope to meet with editor and revisers’ standards. Revised portions are marked with red font in the paper. It mainly includes three parts: simple summary, abstract , introduction and Small part of the discussion.

The main corrections in the responds to the reviewer’s comments are as following.

(1)Line 19 - Explain: "The to protect pigs against ASF has been hindered by."

Answer: We meant that to date, there are still no effective vaccines and antiviral drugs available for the prevention or treatment of ASF. After your reminder, we also found that there is something wrong with this sentence, and it has been deleted from the original text due to a big revision of the abstract.

(2)Line 23- Correct: "102 HAD50 ASFV" and check along the manuscript

Answer: Thanks for your reminder. It was the oversight,"102 HAD50 ASFV" has been changed to "102 HAD50 ASFV" in line 119.

(3)Line 67- A statement on the absence of antivirals and vaccines should be added, as these two references (https://doi.org/10.1080/22221751.2022.2108342; https://doi.org/10.1080/22221751.2021.1902751).

Answer: Thanks you for your suggestion. We haveI read the literaturereferences which you recommended and cited the the relevant references in introduction. The reference was added as Reference 16and 17.

(4)Line 92 - Explain the rational of a single collection and at the 7dpi. Please discuss this limitation of the study in the discussion section.

Answer: Thanks for your question, we have discussed already previously in lines 424 to 435. All pigs infected with 102 HAD50 ASFV CADC_HN09 presented with a rise in body temperature (above 40.5°C) from day 3 dpi, and and quickly evolving to depression, anorexia, staggering gait, diarrhea, and purple skin discoloration.  Included fever (>41 °C) last for 3 days, anorexia, lameness, dyspnea, bloody diarrhea and cyanosis were appeared at 7 dpi, among which 2 pigs died at 7 dpi, and all died at 8dpi. In or-der to compare the most significant differences, microRNA (miRNA) in porcine pe-ripheral blood lymphocytes of ASFV infected pigs and healthy pigs at 7 dpi was com-pared based on Illumina high-throughput sequencing. However, the differences of mi-croRNA (miRNA) at 7 dpi could not fully reflect all the differences caused by ASFV in-fection. Therefore, in future research, the differences of miRNA at different time points, needs to be further investigated and verified, so as to more comprehensively reflect the impact of ASFV infection on the expression of miRNA in host.

(5)Line 166-172: please rephrase the part of the manuscript and remove the description of M&M from here.

Answer: According to your suggestion, we have modified the description in lines 197 to 200 All pigs infected with 102 HAD50 ASFV CADC_HN09 presented with a rise in body temperature (above 40.5°C) from day 3 dpi (Fig. 1), and quickly evolving to full clinical disease (depression, anorexia, staggering gait, diarrhea, and purple skin discoloration), among which 2 pigs died at 7 dpi,and all died at 8dpi.

Finally, we appreciate very much for your time in editing our manuscript and the valuable suggestions and comments. I am looking forward to hearing from your decision when it is made.

With best regards.

Reviewer 3 Report

Dear Authors,

It was a pleasure for me to read your submitted paper, and I feel honored to be one of the reviewers. I think that your results are opening the doors for a deeper understanding of the pathogenicity of ASF, and more... maybe for a better protocol for vaccine development.

However, I do have some suggestions. When I was reading the paper, it felt like I was speaking with 2 different persons. One was talking to me in the first part, and the other in the second part. Please try to rewrite the abstract/introduction to be as professional as the rest of the manuscript.

Here are my point-by-point suggestions

Abstract: Way too long.

Line 13: „Immunity” and „Identified” should be written in lowercase letters

Line 17: „highly contagious hemorrhagic disease” – recent studies which compare the infectiousness of ASF with other diseases, prove that it is not highly contagious, please consult the bibliography better.

Line 19: The sentence “The to protect pigs against ASF has been hindered by.” Has no sense…

Line 20: Replace “interplay” with interaction

Line 23: “then THEY were”

Line 24: “The peripheral blood of pigs infected or uninfected were collected” – What do you mean here? Infected or uninfected? You did not collect from all of them?

Lines 38-40: “Disease and immune pathways implicated include those for leukocyte transendothelial migration, Toll and Imd signaling, influenza A, lysosomes, and endophagy.” – rewrite this phrase, use the correct rules of an enumeration.

Introduction: Treated superficial, if you want to make references regarding the history of ASF, you need to proceed to a better bibliographical study.  You should describe more details regarding miRNA and mRNA, in general, and more specifically about ASFv (even if other studies have negative results)

Line 56: “frames, About 50” – remove the comma, add a point.

Lines 57-58: “ASF first 57 broke out in Kenya in 1921” – false. It was reported for the first time in 1921, but the first outbreak was noticed by the same Montgomery in 1909 in imported big from Europe.

Lines 59-60: “but was completely eradicated from Spain and Portugal by 1995” – not only from those countries, but all western Europe was also ASF clear. The only exception was Sardinia.

Line 64: Ticks… not all of them, only a specific category, please rewrite.

Line 70: “they aretransported to the” – please add the space/break.

-When you use the abbreviation miRNA for the first time, please write also the full term.

Lines 79-81: “Results of this study help to clarify virus-host interactions and provide a theoretical basis for the development of more effective ASFV vaccines that take advantage of the sncRNA system” – maybe the aim was this one.

Materials and Methods: Well-structured. Easy to follow. All protocols are original? If not, please suggest references.

Results

Figure 2: A and B: Please use lowercase or capital for both terms (Infected and control)

Line 202: “These data were reliable and used for subsequent analyses.” – I do not suggest to say like this. Maybe to considered this data to be reliable for your study.

Figure 3: the numbers and words are too small, please adjust the size.

Author Response

Dear Reviewer

Thanks for your letter and for the reviewer’ comments concerning our manuscript entitled “Identification of Potential miRNA–mRNA Regulatory Network Associated with Regulating Immunity and Metabolism in Pigs Induced by ASFV Infection” (animals-2238119). Firstly, we would like to thank the reviewer for those comments which are all valuable and very helpful for revising and improving our paper. We have studied comments carefully and have substantially revised our manuscript which we hope to meet with editor and revisers’ standards. Revised portions are marked with red font in the paper. It mainly includes three parts: simple summary, abstract, introduction and Small part of the discussion.The main corrections in the responds to the reviewer’s comments are as following.

(1)Line 13: „Immunity” and „Identified” should be written in lowercase letters

Answer: Thanks for your discovery. I have corrected this small mistake in line 17.

(2)Line 17: “highly contagious hemorrhagic disease” – recent studies which compare the infectiousness of ASF with other diseases, prove that it is not highly contagious, please consult the bibliography better.

Answer: Thanks for your guidance. We have reviewed the relevant literature and have corrected it. African swine fever (ASF) is a devastating infectious disease in domestic pigs caused by African swine fever virus (ASFV)with mortality rate about 100% in lines 21-22.

(3)Line 19: The sentence “The to protect pigs against ASF has been hindered by.” Has no sense…

Answer: After your reminding, I also found that there is something wrong with this sentence, and it has been deleted from the original text due to a big revision of the abstract.

(4)Line 20: Replace “interplay” with interaction

Answer: It has been modified that however,“the understanding of the interaction between ASFV and host is still not clear”in lines 23.

(5)Line 23: “then THEY were”

Answer: Thanks for your help. We have substantially revised the abstract section accordingly, and the details were also revised.

(6)Line 24: “The peripheral blood of pigs infected or uninfected were collected” – What do you mean here? Infected or uninfected? You did not collect from all of them?

Answer: Thank you for pointing the problem out. The description is really inadequate. In this study,. we meant that we collected samples of all the pigs including the experimental group and control group. Related descriptions have been corrected in the summary in lines 24-26

(7)Lines 38-40: “Disease and immune pathways implicated include those for leukocyte transendothelial migration, Toll and Imd signaling, influenza A, lysosomes, and endophagy.” – rewrite this phrase, use the correct rules of an enumeration.

Answer: Thank you again. We have revised this text again. “The mRNA target genes were enriched into MAPK signaling pathway, Toll-like receptor signaling pathway, TNF signaling pathway and IL-17 signaling pathway by KEGG enrichment analysis”in lines 41-43

(8)Introduction: Treated superficial, if you want to make references regarding the history of ASF, you need to proceed to a better bibliographical study.  You should describe more details regarding miRNA and mRNA, in general, and more specifically about ASFv (even if other studies have negative results)

Answer: Thank you for reminding me. We added some background about miRNA on lines 81-99.

(9)Line 56: “frames, About 50” – remove the comma, add a point.

Answer: Thank you for your help. I have made a large-scale revision of the abstract, and the details there have disappeared from the article.

(10)Lines 57-58: “ASF first broke out in Kenya in 1921” – false. It was reported for the first time in 1921, but the first outbreak was noticed by the same Montgomery in 1909 in imported big from Europe.

Answer: I modified this sentence in the article. You can find it in line 55-56.

(11)Lines 59-60: “but was completely eradicated from Spain and Portugal by 1995” – not only from those countries, but all western Europe was also ASF clear. The only exception was Sardinia.

Answer: I modified this sentence in the article. You can find it in line 57-59.

(12)Line 64: Ticks… not all of them, only a specific category, please rewrite.

Answer: Thank you for your suggestion. Through the modification of introduction, I deleted this expression in the article

(13)Line 70: “they are transported to the” – please add the space/break.

Answer: Thanks for your reminder. This is an error that should not have occurred. I have corrected it. You can find it in line 75.

(14)-When you use the abbreviation miRNA for the first time, please write also the full term.

Answer: Thank you for reminding me. I have modified it.

(15)Lines 79-81: “Results of this study help to clarify virus-host interactions and provide a theoretical basis for the development of more effective ASFV vaccines that take advantage of the sncRNA system” – maybe the aim was this one.

Answer: Yes, I hope the results of this study can provided new basis for further elucidating the interactions between ASFV and host and the immunity regulation mechanisms of ASFV, which will be con-ducive to better controlling ASF.

(16)Materials and Methods: Well-structured. Easy to follow. All protocols are original? If not, please suggest references.

Answer: At present, the method of RNA-seq is very mature. This paper mainly refers to the book “RNA-seq Data Analysis A Practical Approach”.

Results

(17)Figure 2: A and B: Please use lowercase or capital for both terms (Infected and control)

Answer: Thank you for reminding me, it has been modified.

(18)Line 202: “These data were reliable and used for subsequent analyses.” – I do not suggest to say like this. Maybe to considered this data to be reliable for your study.

Answer: Your reminder is correct. I have deleted the relevant argument.

(19)Figure 3: the numbers and words are too small, please adjust the size.

 Answer: I redraw Figure 3, hoping it can meet the requirements.

 Finally, we appreciate very much for your time in editing our manuscript and the valuable suggestions and comments. I am looking forward to hearing from your decision when it is made. With best regards

Round 2

Reviewer 2 Report

The authors had improve the quality of the manuscript by answering to all the requests raised by this reviewer. Thus, in my opinion, the article reaches a suitable form to be published.